# Role of Exosomes in Prostate Cancer Metastasis

**DOI:** 10.3390/ijms22073528

**Published:** 2021-03-29

**Authors:** Theresa Akoto, Sharanjot Saini

**Affiliations:** 1Department of Cellular Biology and Anatomy, Augusta University, Augusta, GA 30912, USA; takoto@augusta.edu; 2Department of Biochemistry and Molecular Biology, Augusta University, Augusta, GA 30912, USA

**Keywords:** exosomes/extracellular vesicles, tumor-derived exosomes, castration-resistance prostate cancer, tumor microenvironment, metastasis

## Abstract

Prostate cancer remains a life-threatening disease among men worldwide. The majority of PCa-related mortality results from metastatic disease that is characterized by metastasis of prostate tumor cells to various distant organs, such as lung, liver, and bone. Bone metastasis is most common in prostate cancer with osteoblastic and osteolytic lesions. The precise mechanisms underlying PCa metastasis are still being delineated. Intercellular communication is a key feature underlying prostate cancer progression and metastasis. There exists local signaling between prostate cancer cells and cells within the primary tumor microenvironment (TME), in addition to long range signaling wherein tumor cells communicate with sites of future metastases to promote the formation of pre-metastatic niches (PMN) to augment the growth of disseminated tumor cells upon metastasis. Over the last decade, exosomes/ extracellular vesicles have been demonstrated to be involved in such signaling. Exosomes are nanosized extracellular vesicles (EVs), between 30 and 150 nm in thickness, that originate and are released from cells after multivesicular bodies (MVB) fuse with the plasma membrane. These vesicles consist of lipid bilayer membrane enclosing a cargo of biomolecules, including proteins, lipids, RNA, and DNA. Exosomes mediate intercellular communication by transferring their cargo to recipient cells to modulate target cellular functions. In this review, we discuss the contribution of exosomes/extracellular vesicles in prostate cancer progression, in pre-metastatic niche establishment, and in organ-specific metastases. In addition, we briefly discuss the clinical significance of exosomes as biomarkers and therapeutic agents.

## 1. Introduction

Prostate cancer (PCa) is the most prevalent cancer among males in United States (US) with an estimated 248,530 cases in 2021 [1]. It remains the second leading cause of male mortality with an estimated 34,130 deaths in 2021 in the US [1]. Serum prostate specific antigen (PSA) is used as a diagnostic biomarker for patients with PCa. However, it has been shown to lack specificity and therefore, lead to PCa overdiagnosis and overtreatment [2,3,4,5]. The survival rate of PCa patients is dependent on early disease diagnosis and available treatment options [6]. This malignancy is regulated by androgens acting via androgen receptor (AR) signaling. Therefore, AR signaling ablation via androgen deprivation therapy (ADT) is the goal of first line of therapies that result in cancer regression initially. However, after a period following hormonal therapy, the tumor progresses to metastatic castration-resistant prostate cancer (mCRPC) [7] with poor survival rates and limited therapeutic options. Hence, the search for new therapeutic regimens continue [8]. 

A majority of PCa-related mortality results from metastatic disease that is characterized by metastasis of prostate tumor cells to various distant organs, such as lung, liver, bone and lymph nodes. Bone metastasis is most common in prostate cancer with osteoblastic and osteolytic lesions [9]. The precise mechanisms underlying PCa metastasis remain unclear which has led scientists to further explore this area. Another area of utmost importance is the identification of alternate prognostic and diagnostic biomarkers which will help in better management of prostate cancer and also be of therapeutic relevance [10]. Some of these biomarkers include circulating tumor cells, microRNAs, and exosomes [5]. 

Exosomes are defined as nanosized extracellular vesicles (EVs), between 30 and 150 nm in thickness, that originate and are released from cells after multivesicular bodies (MVB) fuse with the plasma membrane [11,12]. Exosomes consist of a lipid bilayer membrane enclosing a cargo of biomolecules, including proteins, lipids, RNA, and DNA (Figure 1). Increasing evidence suggests that exosomes mediate intercellular communication by transferring their cargo to recipient cells to modulate target cell functions [13,14]. Intercellular communication is a key feature underlying tumor progression and metastasis. There exists local signaling between tumor cells and various cells within the primary tumor microenvironment (TME), in addition to long range signaling wherein tumor cells communicate with sites of future metastases. This latter signaling promotes the formation of pre-metastatic niches (PMN) that augments growth of disseminated tumor cells upon metastasis in distant organs. In this review, we discuss the contribution of exosomes in prostate cancer progression, in pre-metastatic niche establishment, and in organ-specific (organotropic) metastases. In addition, we briefly discuss the clinical significance of exosomes as biomarkers and therapeutic agents. 

## 2. Exosomes: Heterogeneity and Diverse Roles

Intercellular communication involving the shuttling of biological components such as microRNAs, long non-coding RNAs, DNA, and proteins via exosomes is essential in maintaining cellular homeostasis and normal functioning. However, there are instances whereby the transport of these biological substances across cells via exosomes can be detrimental resulting in pathological conditions. Over the last decade, exosomes have proven to be great export machinery in trafficking biological materials both in normal and diseased states such as in prostate cancer. Exosomes are cell-secreted and widely distributed in different biological or bodily fluids [15,16]. These vesicles consist of lipid bilayer membrane composed of cholesterol, phosphatidylserines, and sphingolipids which protect them against proteases and RNases [17,18]. Exosomes play key roles in different physiological processes, examples including erythrocyte maturation from reticulocytes, immune cell responses and also spreading of pathogens (virus and prions) from cell-to-cell [19,20]. In cancer, tumor-derived exosomes lead to malignancy, enhanced cell survival and proliferation, drug resistance and the creation of a PMN. The shuttling of the biological components contained in exosomes is thought to be a contributing factor to cancer development. According to Exocarta, a database for collating exosomal content, 9769 proteins, 3408 mRNAs, 2838 microRNAs and 1116 lipids have been identified in exosomes suggesting their crucial roles in intercellular communication [21,22,23,24]. Here we highlight some of the specific constituents in exosomes (Figure 1) and their key roles and functions. Also shown in Table 1 is a summary of exosomal markers that are specific to PCa and their roles. 

### 2.1. Exosome-Contained Proteins

The protein content of exosomes varies based on the cell types and bodily fluids from which they are produced [25]. They are typically purified and analyzed using Western blotting, fluorescence-activated cell sorting (FACS), mass spectrometry, and proteomics [26,27]. The endosomal sorting complex required for transport (ESCRT) consisting of four cytosolic protein complexes (ESCRT-0, ESCRT-I, ESCRT-II, and ESCRT-III) with about twenty proteins collectively including their accessory proteins such as programmed cell death 6-interacting protein (Alix), Tumor susceptibility gene 101 (TSG101), Vacuolar protein sorting 4 (VPS4), Vps twenty-associated-1 (VTA1), Heat shock cognate 71 kDa protein (HSC70), and Heat shock protein 90 β (HSP90β). In addition to these proteins, Ras-associated binding (Rab) family of proteins are major proteins in exosomes which contribute to exosomal biogenesis and serve as protein markers in identifying these exosomal vesicles [6,28,29]. Exosomes also consist of specific molecular components derived from their original cells which include transmembrane markers in the tetraspanin family (CD81, CD82, CD83, CD9, and CD63) [19,26,30,31]. These tetraspanins play diverse roles in biological processes such as cell adhesion, membrane fusion, motility, invasion, and exosome regulation [31,32,33]. Further, exosomes are enriched in other proteins such as cell adhesion molecules (e.g., integrins), growth factor receptors, phosphatidylserine-binding milk-fat globule-epidermal growth factor 8 (MFG-E8)/lactadherin and heterotrimeric G proteins [34,35,36]. Geis et al. reported that myeloid-derived suppressor cells (MDSC)-derived exosomes carry proteins, in addition to mRNAs and microRNAs (miRNAs) with some exosomes having different constituents than those of their parental cells [37]. They found that proteins such as annexins (A4, A6, and A7), histones (H2A.z, H2B, H3.1, H3.2, and H4), CD9, heat shock proteins (Hsp70 protein-4, Hsp cognate 71 kDa, heat shock protein 90α (Hsp90α), Hsp90β, major histocompatibility complex II (MHC II) molecules, Alix and vacuolar protein sorting-associated proteins Vps25, Vps4B, and Vps37B were enriched in exosomes with proteins CD9, Vps4B, and Vps37B only identified in exosomes in comparison with their parental cells [37,38]. In addition, proinflammatory proteins; S100A8 and S100A9 loaded in exosomes from MDSCs contributed to the immunosuppressive role of MDSC and enhanced tumor growth and metastasis [37,38]. In the tumor microenvironment, cancer stem cells and mesenchymal cells play integral roles in tumor progression [39]. Proteins loaded in exosomes derived from these two cell types have been shown to serve as intermediates of crosstalk between non-cancer stem cells and cancer stem cells thereby contributing to differentiation and dedifferentiation between these two cell types, respectively [39]. Moreover, analysis from Exocarta database revealed that exosomes from mesenchymal stem cells were enriched in proteins such as cytokines (LIF, CSF1–3, FAM3B), chemokines (CCL-2, -7, -20, and -28, CXCL -2 and -16), interleukins (IL-3, -5, -7, -9, -10, -17B, -19, -23A, -36G, OSM), and growth factors (VEGF, HGF, ITGB1, TGFB1, FGF18, PDGFA, TGFB2, ESM1, INHBB, FGF19, NRG2, GDF5, GDF3, INHBA, FGF16, GDF11, PDGFC, GDF1, GDF9) which may be involved in signaling [21,22,23,24,34]. Finally, exosomes are also enriched in cytoplasmic enzymes (pyruvate kinases, GAPDH, peroxidases, and lactate dehydrogenase) as well as cytoskeletal proteins (tubulin, actin, myosin, cofilin 1 and ezrin-radixin-moesin (ERM) proteins) [40,41]. 

In PCa, numerous studies have reported proteins that are selectively packaged into exosomes for different cellular and biological functions. Hosseini-Beheshti et al. identified several proteins that were enriched in exosomes in different PCa cells after proteomics profiling [42]. In their study, the authors reported that exosomes from RWPE-1; immortalized normal prostate cell line contained proteins such as Inter-Alpha-Trypsin Inhibitor Heavy Chain 4 (ITIH4), Laminin Subunit Gamma 2 (LAMC2), hemoproteins (HEMO), Keratin, Type II Cytoskeletal 6A (K2C6A), FAT Atypical Cadherin 2 (FAT2), Keratin, type II cytoskeletal 5 (K2C5), Keratin, type I cytoskeletal 14 (K1C14), and Small Heat Shock Protein Beta-1 (HSPB1) [42]. In addition, they identified exosomal proteins such as Endoplasmin (ENPL), Glucose-Regulated Protein, 78kD (GRP78), and Annexin A2-like protein (AXA2L) to be specific in AR negative PCa cells; DU145 and PC3 cells [42]. These proteins may contribute to the understanding of PCa and provide therapeutic targets in the management of the condition. Further, proteomic analyses of exosomes from PCa cell lines (PC346C and VCaP) and immortalized primary prostate epithelial cells (PNT2C2 and RWPE-1) revealed about 481 and 385 proteins respectively including Fatty acid synthase (FASN), Programmed Cell Death 6 Interacting Protein (PDCD6IP), Exportin 1 (XPO1) and Enolase 1 (ENO1) which could serve as biomarkers of PCa [43]. To identify exosome-rich proteins that contribute to the high Gleason Score in 18 men using liquid chromatography-tandem mass spectrometry, Fujiti et al. identified about 4710 proteins contained in exosomes isolated from urine from the patients [44]. Interestingly, among these lists of proteins, proteins such Charged multivesicular body protein 4a (CHMP4A), Charged multivesicular body protein 4c (CHMP4C), and Charged multivesicular body protein 2b (CHMP2B) which are components of ESCRT-III alongside proteins such as Granulin Precursor (GRN) and Alpha-1 microglycoprotein (AMBP) were found to be overexpressed in men with a Gleason Score of 8–9 suggesting that these proteins may con-tribute to dramatic PCa progression and metastasis [44]. Also, mass spectrometry analyses by Ishizuya et al. revealed 787 exosomal proteins, including Alpha-2-Macroglobulin (A2M), Immunoglobulin Heavy Constant Mu (IGHM), Calcium Release Activated Channel Regulator 2A (CRACR2A), Complement C1q Tumor Necrosis Factor-Related Protein 3 (C1QTNF3), CAMP Responsive Element Binding Protein 3 (CREB3), and Upstream Binding Transcription Factor (UBTF), from sera of 36 men with castration-resistant prostate cancer (CRPC), which may be contributing to PCa disease progression to CRPC [45].

### 2.2. Exosome-Contained MicroRNAs, Messenger RNAs, and Long Non-Coding RNAs

MicroRNAs (miRNAs), messenger RNAs (mRNAs), and long non-coding RNAs (lncRNAs) are essential components of exosomes involved in various biological functions in a variety of cancers [46]. miRNAs can play oncogenic and tumor-suppressive roles in cancer development underlying the importance of their study [47]. Exosomal miRNAs may communicate via gap junctions with neighboring cells of the same tissue or adjacent tissues [48]. He et al. demonstrated that exosomal miR-205 is a positive regulator of ovarian cancer metastasis [49]. By examining the expression of circulating miR-205 in the serum of ovarian cancer patients using the GEO database (GSE106817), the authors found that circulating exosomal miR-205 from serum of patients contributed to ovarian cancer metastasis [49]. They further found that exosomal miR-205 regulates angiogenesis in vitro and in vivo via regulation of PTEN/AKT signaling [49]. Teng et al. reported high levels of tumor suppressor miR-193a in exosomes from colon cancer patients [50]. The authors delineated that, miR-193a interacted with major vault protein (MVP) in cells and was selectively packaged into exosomes leading to colon cancer metastasis to the liver [50]. In another study, exosomal miR-21 and miR-29a were shown to contribute to tumor growth and metastasis by activating toll-like receptors 7 and 8 in immune cells thereby activating a premetastatic inflammatory response [51,52]. Examples of some miRNAs contained in exosomes in prostate cancer include miR-409 and miR-141 which are upregulated and can lead to epithelial-mesenchymal transition (EMT), tumorigenesis, bone metastasis, and AR activation by targeting small heterodimer partner (SHP) [48,53,54]. Exosomal miR-141 and miR-375 were also found to be present in serum of metastatic PCa patients [55]. Furthermore, exosomal miR-105 secreted by PCa, breast, ovarian, and gastric cancers was shown to enhance tumorigenesis and promote metastasis and invasion [56,57].

Exosomal mRNA composition is slightly different from donor cells due to the absence of ribosomal RNAs (18S and 28S) in them [14,57]. Valadi et al. reported that exosomes derived from mouse mast cell line, human mast cell line, and bone marrow-derived mouse mast cells (MC/9, HMC-1, BMMC respectively) contain functional mRNAs and microRNAs (let-7, miR-1, miR-15, miR-16,miR-17, miR-18, miR-181, and miR-375) [14,40]. The mRNAs contained in the exosomes derived from MC/9 coded for Cytochrome c oxidase subunit 5B (Cox5b), Heat shock 70 kDa protein 8 (Hspa8), Serine hydroxymethyltransferase 1 (Shmt1), Lactate dehydrogenase 1 (Ldh1), Zinc finger protein 125 (Zfp125), glucose-6-phosphate isomerase 1 (Gpi1) and Rad 23 homolog B nucleotide excision repair protein (Rad23b) proteins confirming their functional protein-coding abilities [14]. In addition, using Ingenuity Pathway Analysis, the mRNAs were identified to be significant in pathways contributing to cellular development, RNA post-translational modifications, protein synthesis and may also be mediating transport and communication across cells [14]. Also, specific exosomal miR-1, miR-17, miR-18, miR-181, and miR-375 from their study played diverse roles and functions including angiogenesis and tumorigenesis [14]. In a different study, the serum of patients with esophageal cancer contained high levels of esophageal cancer-related gene-4 (ECRG4) mRNA, a tumor suppressor gene that was found to inhibit tumor growth and angiogenesis [58]. Nilsson et al. showed the presence of mRNAs; prostate cancer antigen 3 (PCA3) and transmembrane protease serine 2:v-ets erythroblastosis virus E26 oncogene homolog (TMPRSS2:ERG) in tumor exosomes from the urine of PCa patients [59]. PCA3 gene is a segment of noncoding mRNA located on chromosome 9q21–22 that is expressed significantly, about 60–100 folds in PCa tissues as compared with normal tissues [60,61]. TMPRSS2:ERG is the predominant gene fusion in PCa where TMPRSS2 protein functions with overexpressed ERG; an ETS transcription factor thereby contributing to androgen-resistance in PCa [62,63,64]. A recent study showed that AR variant, AR-V7 RNA is present in exosomes in patients treated with AR pathway inhibitors Enzalutamide and Abiraterone as a primary therapy and is predictive of resistance to AR signaling inhibitors [65].

Over the past decade, lncRNAs have gained attention in exosome research. LncRNAs are typically about >200 nucleotides in length and do not code for proteins however, they are able to interact with proteins, DNAs, and RNAs in the nucleus or cytoplasm [66,67]. Exosomal lncRNAs have been implicated in angiogenesis, tumor growth and migration, and reprogramming the TME [68,69,70]. Several studies have reported lncRNAs to be selectively packaged in exosomes for their biological functioning in recipient cells and could serve as biomarkers for various cancers such as PTENP1 (bladder cancer), MALAT1 (non-small cell lung cancer and breast cancer), HOTAIR (laryngeal squamous cell carcinoma), GAS5 (colorectal cancer), TUG1 and CCND1(cervical cancer and breast cancer) CCAT2 and POU3F3 (glioma), lnc-ROR, lnc-ROR, lnc TUC339 and lnc VLDLR (hepatocellular carcinoma) [40,67,69,70,71,72,73]. In prostate cancer, exosomal lncRNA MYU and PCSEAT function to enhance cell proliferation and migration, whereas exosomal lincRNA-p21, SAP30L-AS1, SChLAP1 function as diagnostic biomarkers [74,75,76,77].

### 2.3. Exosome-Contained Lipids

Lipids serve as very essential components of the plasma membrane of exosomes contributing to exosomes’ biogenesis, release, and function [78]. Exosomes purified from Oli-neu cells, a mouse oligodendroglial cell line were enriched in ceramides which triggered multivesicular endosome formation from endosome vesicles [79]. Further, exosomes isolated from B-lymphocytes contained acidic lipids (ganglioside GM3 with palmitoyl or nervonoyl residues, phosphatidylinositol, phosphatidylserine, and phosphatidic acid), and uncharged lipids (sphingomyelin, cholesterol, phosphatidylcholine, phosphatidylethanolamine, and ether lipids with an ethanolamine phosphoryl head group) [80]. Also, exosomes collected from two myeloid cell types; rat mast cells (RBL-2H3 mast cell line) and human dendritic cells and their parent cells showed similar lipid distribution; high sphingomyelin, lysophosphatidylcholine, and phosphatidylethanolamines suggesting a rigid lipid packaging in these exosomal membranes than their parent cells [81]. In addition, exosomes derived from reticulocytes were enriched in sphingomyelin than their parent cells although studies by Carayon et al. reported variations in lipid composition in a seven-day in vitro reticulocyte maturation experiment. This shows the participation of these lipids in different molecular mechanisms in exosome packaging during reticulocyte maturation into erythrocytes [11,82]. Finally, exosomes can also serve as carriers in transporting bioactive lipids (fatty acids, prostaglandins, and leukotrienes) and lipid metabolism enzymes (phospholipase A2, C, and D) to recipient cells [40,83,84]. In prostate cancer, Llorente et al. employed targeted lipidomic assays and reported 217 lipid species from exosomes isolated from PC3 cells [85]. They observed that exosomes were highly enriched in glycosphingolipids, cholesterol, sphingomyelin, and phosphatidylserine (1.2%, 43.5%, 16.3%, and 31.1%, respectively) as compared to their parent PC3 cells (0.5%, 19.2%, 6.9%, and 7.7%, respectively) suggesting that these lipids may play major roles in exosome regulation and could be potentially used as exosome biomarkers [85].

### 2.4. Exosome-Contained DNAs

Studies have shown that tumor-derived exosomes carry DNA. These exosomal DNAs may be used to classify mutations that exist in parental tumor cells [86,87]. In research conducted by Thakur et al., the authors reported the evidence of double-stranded DNA (dsDNA) enriched in tumor-derived exosomes specifically in melanoma, breast, lung, prostate, and pancreatic cancers [87]. Although the aforementioned tumors contained dsDNA, exosomes collected from pancreatic and lung cancer cell lines were lower in DNA content compared with the other cancers suggesting the variations in exosomal DNA packaging in different cancer cells [87]. The authors also showed exosomal DNA could carry *BRAF* mutations, common mutations in malignant melanoma, and mutations in epidermal growth factor receptor (*EGFR*) in NSCLC [87]. Performing allele-specific polymerase chain reaction (AS-PCR) on primary melanoma cell lines and NSCLC cell lines to characterize mutational status in *BRAF* and *EGFR*, respectively in exosomal DNA, their findings portrayed that exosomal DNA could be used to classify the mutational status of these two genes in parental cell lines [87]. Finally, circulating exosomal DNAs from serum of pancreatic cancer patients and healthy patients were identified to contain homozygous *TP53* mutation at codon 273 and heterozygous *KRAS* mutation at codon 12 [88]. In prostate cancer, studies show that tumor genomic alterations, including copy number variations of genes frequently altered in metastatic PCa (i.e., *MYC*, *AKT1*, *PTK2*, *KLF10,* and *PTEN*), are represented in large EVs [86]. 

## 3. Tumor-Derived Exosomes: Role in Prostate Cancer Progression and Metastasis

Tumor-derived exosomes have been implicated in prostate cancer progression and metastasis to bone and other sites. Intercellular communication mediated by exosomes play an integral role in mediating both short-range and long-range communication between cancer cells, tumor microenvironment and metastatic sites. The following sections highlight the role of exosomes in these processes in prostate cancer. 

### 3.1. Role of Exosome-Mediated Communication in Prostate Tumor Microenvironment and Angiogenesis

For cancer cells to maintain their hallmarks; sustain immortality, continue their proliferative activity, evade immune responses, resist cell death or apoptosis, initiate angiogenesis, continue invasion and metastasis, a medium is required [102,103,104]. Tumor microenvironment (TME) serves as this medium and it is composed of different cell types such as stromal cells (pericytes, fibroblasts, mesenchymal stromal cells, neuroendocrine cells, etc.), immune cells (T and B lymphocytes, dendritic cells, macrophages, etc.) and the extracellular matrix (ECM) which interact with the tumor cells to enhance their survival and growth. Tumor-derived exosomes from cells in the TME function as important participants or signals in maintaining cell-cell activity [17,105,106]. Tumor-derived exosomes in TME have been implicated as contributing factors in cancer therapy resistance mainly due to their involvement in the myriad of signaling activities across the different cell types in the TME, giving the tumor cells the ability to adapt and evade treatment [107,108]. Tumor-derived exosomes affect the ECM by disrupting cell adhesion via integrins and degrading collagen and fibronectin resulting in enhanced invasion and metastasis [109,110]. Furthermore, studies by DeRita et al. showed that prostate cancer cell exosomes contain the protein Src which activates focal adhesion kinase through integrin activation leading to angiogenesis and metastasis [111]. Also, studies showed that exosomes derived from prostate cancer cells enhance angiogenesis by promoting matrix metalloproteinases, under hypoxic conditions which leads to vascular leakiness and promotes circulating tumor cells (CTC) invasion [112,113,114]. 

### 3.2. Tumor-Derived Exosome Mediated Communication in Pre-Metastatic Niche Formation and Organ-Specific Metastasis

Tumor-derived exosomes have been implicated in tumor growth, invasion and metastasis by communicating with cells at distant, pre-metastatic organ sites via the formation of tumor favoring microenvironments, a process referred to as ‘pre-metastatic niche’ formation, in addition to their role at metastatic sites [115] (Figure 2). The formation of PMN involves persistent long-range intercellular communication mediated by soluble or membrane-bound factors originating from the primary tumor [113,116]. These factors traverse through vascular and interstitial spaces and eventually engage with the cholesterol-rich plasma membrane of recipient cells at pre-metastatic organ sites [117]. Tumor-derived exosomes remodel the ECM by accumulating fibronectin and promoting ECM-modifying enzyme lysyl oxidase (LOX) crosslinking to enhance the adhesion of the bone-marrow-derived cells which are essential pre-metastatic niche components [113,118,119]. Interestingly, a recent study elegantly demonstrates that EV-mediated communication between PCa cells and bone marrow is mediated by cholesterol homeostasis in bone marrow myeloid cells [117]. They showed that PCa EV uptake by bone marrow myeloid cells in vitro and in vivo leads to activated NF-κB signaling, reduced myeloid thrombospondin-1 expression and enhanced osteoclast differentiation. A reduction of myeloid cell cholesterol in vitro and in vivo via a targeted, biomimetic approach prior to conditioning with PCa EVs prevented the uptake of PCa EVs by recipient myeloid cells, abolished NF-κB activity, decreased osteoclast differentiation, stabilized thrombospondin-1 expression and led to reduction of metastatic burden by 77% [117]. In another study, the authors demonstrate a role for phospholipase D2 in stimulating exosome secretion in PCa cell line models as well as a potential to increase osteoblast activity, thereby promoting the establishment of PCa bone metastasis [120]. 

Epithelial-to-mesenchymal transition (EMT) is a key step in the initiation of metastasis which is activated by autocrine and paracrine signaling by tumor-derived exosomes which can then target EMT-related factors such as β-catenin and transforming growth factor-beta (TGF-β) [121,122,123]. Prostate cancer-derived exosomes from DU145 and PC3 cells produce TGF-β capable of inducing transformation of fibroblasts into myofibroblasts by triggering the TGF-β/SMAD signaling pathway and also promoting tumorigenesis and suppressing immune response [124,125]. Exosomes secreted by prostate tumor cells contain a protein hyaluronidase 1 (Hyal1) which promotes the migration of prostate stromal cells; in effect enhancing prostate cancer progression [126]. Studies by Josson et al. have shown that stromal-derived exosomes contain miR-409, which contribute significantly to EMT in prostate cancer [54]. Similarly, exosomes collected from LNCaP and PC3 cells contained Integrin Subunit Alpha 3 (ITGA3) and Integrin Subunit Beta 1 (ITGB1) proteins which contribute to the migration and invasion of these tumors [127]. Interestingly, a study shows that miR-125b, miR-130b, and miR-155 derived from exosomes trigger neoplastic transformation and mesenchymal-epithelial transition (MET) of stem cells derived from adipose tissues of patients with prostate cancer [128]. miR-105 from tumor-derived exosomes functions as a tumor suppressor and prevents cell proliferation in prostate cancer by targeting cyclin-dependent kinase 6 (CDK6) [56,129]. 

### 3.3. Role of Tumor-Derived Exosomes in PCa Lymphatic Metastasis

The lymphatic system is made up of the lymphatic vessels and lymphoid organs comprising of lymphocytes, lymph nodes, spleen, and tonsils. Over the years, a strong association has been found between tumor cells and tumor metastasis particularly in PCa [130,131]. An interesting route PCa-derived cells adapt during metastasis is through the lymphatic route where tumor cells intravasate into lymph nodes to get to their target sites thereby promoting aggressive phenotypes in PCa [131,132,133]. Therefore, research is gearing towards understanding how these PCa cells can metastasize into the lymph nodes. Although mechanisms by which tumor cells initiate this process seems to be a conundrum, one likely mediator contributing to lymph node metastasis to distant organs may possibly be via exosomes [134,135,136,137]. In a study by Khan et al. using tumor-derived samples from PCa patients, the authors identified significant expression of Survivin; a protein classified under inhibitor of apoptosis (IAP) family in exosomes collected from these patients [138]. Interestingly, increase in expression of Survivin was also identified in lymph node tissues from prostate carcinoma patients with high Gleason Score and was found to contribute to metastasis to lymph nodes and aggressive PCa behavior [139]. In another study by Maolake et al., the authors divulged that by activating Chemokine (C-C motif) ligand 21/C-C chemokine receptor 7 (CCL21/CCR7) axis, Tumor necrosis factor-α (TNF-α) contributed to PCa lymph node metastasis using DU145, PC3, LNCaP-SF, and LNCaP prostate cancer cell lines [140]. TNF-α has been reported to be packaged in exosomes [141] and this observation may mediate their effect shown in lymphatic metastasis. Further, Gaballa et al. recently demonstrated that integrin alpha 2 subunit (ITGA2) was packaged in exosomes isolated from DU145, PC3, LNCaP, and CWR-R1 PCa cells which are all metastatic PCa cell lines [142]. Inhibition of exosome release to recipient cells by knocking down ITGA2 in these metastatic PCa cells suggested the potential role of ITGA2 in contributing to disease progression and aggressive phenotypes observed in PCa [142]. In addition, although there were no significant changes in the protein expression levels of ITGA2 in 24 primary PCa tissues verses their matched metastatic lymph node tissues, lymph node metastatic tissues with a high Gleason Score of 9 was observed to have increased expression of ITGA2 in comparison with PCa tissues with lower Gleason Score of 7, indicating the likely role of ITGA2 in enhancing lymph node metastasis [142].

### 3.4. Role of Tumor-Derived Exosomes in Drug Resistance in Prostate Cancer

Chemotherapy remains one of the classical ways of treating advanced PCa [143]. However, several factors including tumor heterogeneity [144], epigenetic regulation by miRNAs [31], synergistic effects of multiple signaling pathways such as Akt/PI3K, MAPK/ERK, NF-κB/IL-6, mTOR, Hedgehog, and somatostatin receptor have been shown to contribute to drug resistance in these tumor cells. Thus, drug resistance has been one of the most daunting challenges in prostate cancer chemotherapy. The TME has also been implicated as one of the drivers of drug resistance in PCa [145] and the role of tumor-derived exosomes in modulating the TME remains an interesting area of study. We recently demonstrated that enzalutamide resistance and induction of therapy-induced neuroendocrine differentiation states in prostate cancer is mediated via exosomes [146]. Inhibition of exosome release could partly revert the sensitivity of enzalutamide resistant prostate cancer cells. Further, upon enzalutamide treatment, neural transcription factors BRN2 and BRN4 were released in prostate cancer exosomes that drive oncogenic reprogramming of prostate adenocarcinomas towards neuroendocrine states [146]. Corcoran et al. reported that exosomes contributed to docetaxel resistance in docetaxel-sensitive PCa cells (DU145, 22Rv1, and LNCaP) [147]. Release of multidrug resistance protein 1/P-glycoprotein (MDR-1/P-gp), a transporter protein that is associated with the efflux of many foreign substances including anti-cancer drugs from exosomes, was a likely factor in conferring resistance to the docetaxel-sensitive cells after the authors treated the cells with exosomes obtained from docetaxel-resistant variants of DU145 and 22Rv1 (DU145RD and 22Rv1RD), respectively [147]. Similarly, studies from Kato and colleagues demonstrated that MDR-1/P-gp released by serum exosomes in CRPC patients could be a marker for causing docetaxel resistance in PC3 cells [99]. Interestingly, knockdown of exosomal P-gp could decrease docetaxel resistance in the PC3 cells [99]. In addition, using nanoparticle tracking analysis, Kharaziha et al. revealed that docetaxel-resistant DU145 PCa cells released increased amount of exosomes as compared with docetaxel-sensitive DU145 PCa cells [148].

## 4. Clinical Utility of Exosomes as Biomarkers in Prostate Cancer Management

Over the years, exosomes and other EVs such as oncosomes, apoptotic bodies, microvesicles, and prostasomes are growing as rich sources of biomarkers for cancer diagnosis and as sources of tumor-derived proteins for cancer development [149,150,151]. Due to the involvement of exosomes in an array of biological signaling pathways and acting as cargoes in the tumor microenvironment, they can serve as an attractive source of biomarkers for PCa management. Exosomes are cell-secreted and detectable in biological fluids such as saliva, urine, serum, semen, prostatic secretions, etc., and can act as biomarkers for cancer detection and monitoring of response to therapies [20,43,151,152,153].

In a study conducted by Huang et al., profiling of plasma exosomal miRNA in CRPC patients identified exosomal miR-375 and miR-1290 as potential prognostic biomarkers for predicting survival in CRPC cases [89]. In identifying exosomal biomarkers, Bhagirath et al. reported exosomal miR-1246 as a potential exosomal biomarker in distinguishing between normal, benign, and aggressive form of PCa by showing a 31- and 23-fold increase in aggressive PCa in comparison with normal and benign prostatic hyperplasia (BPH), respectively [96]. Treatment of PC3 and LNCaP cells with exosomal inhibitor GW4869 further confirmed miR-1246 to be released specifically in PCa exosomes [96]. In addition, overexpression of miR-1246 in PCa xenograft mouse model in vivo highlighted the tumor-suppressive role of this miRNA suggesting its therapeutic relevance in managing aggressive PCa [96]. We recently reported exosomal thrombospondin 1 as a potential biomarker for detecting treatment-induced neuroendocrine differentiation (NED) in mCRPC patients (Bhagirath et al., Scientific Reports, under revision). Concurrently, after identifying myriad of altered miRNAs in CRPC-Neuroendocrine (CRPC-NE) and CRPC-Adenocarcinomas (CRPC-Adeno) exosomes/EVs, we used machine learning algorithms to next-generation sequencing miRNA dataset and identified a novel ‘EV-miRNA classifier’ consisting of miR-28-5p and miR-148a-3p that could have clinical advantage in distinguishing between CRPC-NE and CRPC-Adeno in patients non-invasively (Bhagirath et al., Scientific Reports, under revision). Since aberrant androgen receptor signaling can contribute to the development of resistance in PCa cells by expressing truncated AR variant 7 (AR-V7) proteins, detecting AR-V7 in exosomes, CTCs, and circulating tumor RNA (ctRNA) from PCa patients has proven to serve as a potential biomarker in CRPC. In a study conducted by Foj et al. [154], urinary exosomes containing miR-21 and miR-375 from 52 patients were identified as phenomenal non-invasive prognostic biomarkers in managing PCa. Following up on proteomics data of LNCaP- and PC3-exosomes from Sardana et al., Bijnsdorp et al. reported an increase in exosomal proteins ITGA3 and ITGB1 in urine of metastatic PCa patients which could serve as diagnostic metastasis biomarkers [127]. This study showed that exosomal ITGA3 and ITGB1 may play a role in manipulating non-cancerous surrounding cells [127]. Exosome-associated BRN2 and BRN4 were found to possess diagnostic potential for therapy-induced neuroendocrine prostate cancer [146]. Overall, preceding studies highlight the clinical significance of exosomes as prostate cancer biomarkers. Further, exosomes are also efficient and effective carriers of drugs that are used in PCa treatment [15,16].

## 5. Exosomes as Drug Carriers in Prostate Cancer Management

Chemotherapy remains one of the exacting tasks for researchers in the field especially due to their low bioavailability, drug resistance, immune response threats, and lack of specificity of the drugs used for treatment [155]. The use of nanotechnology has been explored by various scientists as a promising tool in the administration of these chemotherapeutic drugs to their target sites [155,156]. Several advantages accompany the use of these nanoparticles, not to mention but a few, including the protection of cargo from immune response, faster delivery and increase in its bioavailability, although there have been some drawbacks in their usage such as low specificity to tumors, and the induction of chemoresistance and radioresistance in tumor cells [155,157,158]. Nevertheless, due to their nanosized shape and multiple biological functions, exosomes have proven to be excellent carriers of miRNA, siRNA, mRNA, lncRNA, peptides, and synthetic drugs to target sites and this makes them interesting tools for use in cancer chemotherapy [159]. In loading these synthetic and small molecules into exosomes, studies by Batracova et al. showed three possible mechanisms—(1) naïve exosomes collected from parental cells can be loaded with cargoes ex vitro; (2) parental cells can be loaded with the cargoes which are then released into exosomes, and lastly (3) using transfection whereby parental cells absorb DNA encoding therapeutically active compounds which are then released in exosomes [160]. Kosaka et al. demonstrated that tumor suppressor miRNA, miR-143 from non-cancerous cells was delivered by exosomes to PCa cell line; PC-3M-luc cells thus inhibiting proliferation of the tumor cells in vitro and in vivo [161]. Also, extracellular vesicles (exosomes and microvesicles) derived from LNCaP and PC3 prostate cancer cell lines could serve as efficient carriers in delivering Paclitaxel, a drug for managing metastatic PCa to their parental cells by increasing its cytotoxic effect [16,162]. However, one major challenge observed by the authors was that EVs without the drug could increase cell viability [162].

## 6. Conclusions and Future Studies

Numerous studies conducted have shown that exosomes contribute to the cell-to-cell interactions within the tumor microenvironment, support pre-metastatic niche development and distant metastasis. However, several questions need to be addressed in future studies. Outstanding questions include: What exosome biogenesis pathways are active in prostate cancer cells that can eventually be targeted for therapeutic applications? How distinct are exosomal contents from primary prostate cancer versus metastatic prostate cancer? Do the primary prostate tumors continue supporting the metastatic tumors and vice-versa? It has been shown in prostate cancer and other tumors that pre-metastatic niche formation is mediated by exosomes, driven by continuous exchange of proteins and microRNAs in exosomes [115]. Future studies are needed to examine if pre-metastatic niche formation in PCa is reversible and if it can be exploited for therapeutic targeting of PCa metastasis. Also, it needs to be examined if exosomes from PCa metastatic sites favor development of further pre-metastatic niches in other distant organs like other tumor types [115]. Seeking answers to these outstanding questions will not only increase our understanding of exosomes in prostate cancer biology but will also pave the way for novel therapies for advanced prostate cancer that are urgently needed. Also, increased understanding will promote the optimized development of exosomes as alternate disease biomarkers, particularly for aggressive, metastatic prostate cancer.

## Figures and Tables

**Figure 1 ijms-22-03528-f001:**
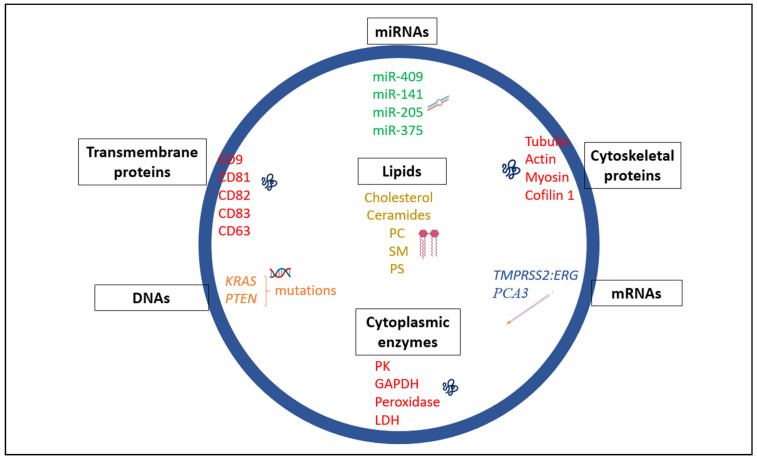
Schematic of exosome heterogeneity. Exosomes typically serve as vehicles in shuttling a cargoes of proteins, miRNAs, mRNAs, DNAs, and lipids across cells. Phosphatidylcholine is abbreviated as PC, sphingomyelin as SM, phosphatidylserine as PS, Pyruvate Kinase as PK, lactate dehydrogenase as LDH, and glyceraldehyde 3-phosphate dehydrogenase as GAPDH.

**Figure 2 ijms-22-03528-f002:**
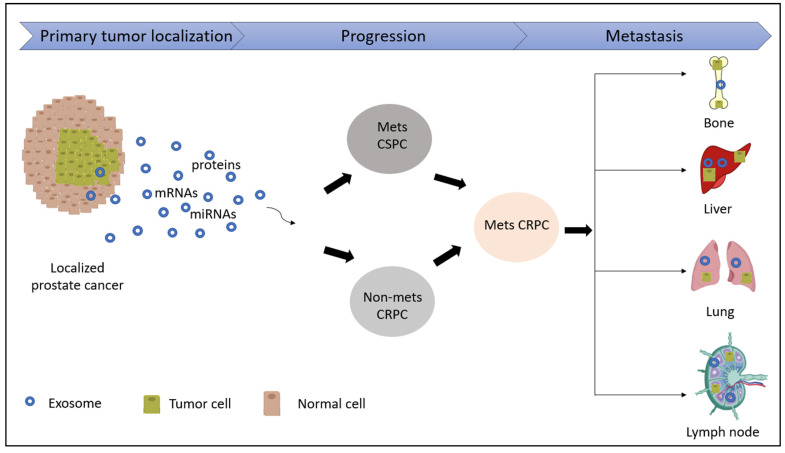
Schematic illustrating potential role of exosomes in prostate cancer progression. The release of tumor-derived exosomes enhances crosstalk across cells, leading to the progression of a localized PCa into either a metastatic Castration-Sensitive PCa (Mets CSPC) or a non-metastatic Castration-Resistant PCa (Non-mets CRPC). This ultimately leads to the advanced form of the disease; metastatic Castration-Resistant PCa (mCRPC) which can metastasize into various organs of the body such as bone, liver, lungs and lymph nodes.

**Table 1 ijms-22-03528-t001:** Exosomal Markers in Prostate Cancer.

Exosome Source(s)	Isolation Method	Component(s) in Exosome	Type	Application(s)	Role in PCa	Reference(s)
Serum	ExoMiR extraction kit	miR-375, miR-141	miRNA	Diagnostic markers	Enhances tumor progression	[55]
Plasma	ExoQuick-based assay	miR-1290, miR-375	miRNA	Prognostic markers	Contributes in the progression to CRPC	[89]
Urine	Ultracentrifugation	TM256,LAMTOR1, VATL, ADIRF	protein	Diagnostic markers	Upregulated in PCa; may contribute to disease progression	[90]
Urine	EXOPRO Urine Clinical Sample Concentrator Kit	*ERG, PCA3,SPDEF*	mRNA	Diagnostic markers	Promotes PCa aggressive phenotype	[91]
Plasma/serum	Differential ultracentrifugation	αvβ3 integrin	protein	Prognostic/diagnostic marker	Promotes PCa aggressive phenotype	[92]
Urine	Differential centrifugation	* PCA3,TMPRSS2:ERG *	mRNA	Diagnostic markers	Regulate AR signaling, mediate oncogenic signaling	[59,93]
Urine	Differential centrifugation	miR-196a-5p, miR-501-3p, miR34a-5p, miR-92a-1-5p, miR-143-3p	miRNA	Diagnostic markers	Downregulated in PCa	[94]
Urine	Differential centrifugation	miR-574-3p, miR-141-5p, miR-21-5p	miRNA	Diagnostic markers	Upregulated in PCa	[95]
Serum	Total Exosome Isolation Reagent	miR-1246	miRNA	Diagnostic marker	Predicts aggressiveness in PCa	[96]
Bone metastasis	Ultracentrifugation	Annexins A1, A3 and A5, DDAH1	proteins	Predictive markers	Contributes to angiogenesis in PCa	[97]
VCaP, LNCaP, C4–2 PCa cells	Ultracentrifugation	CLSTN1, FASN, LDHA	proteins	Diagnostic and prognostic markers	Enhances PCa progression	[42]
DU145, PC3 cells	Ultracentrifugation	ANXA2, CLU, ENO1, FN1, KRT8, LAMA5, NPM1, PRDX1, TFRC	proteins	Diagnostic and prognostic markers	Enhances PCa progression	[42]
DU145, PC3, VCaP, LNCaP, C4–2 prostate cancer cells, RWPE-1 benign prostate cells	Ultracentrifugation	Sphingolipid, glycosphingolipid	lipids	predictive markers	Enhances PCa progression	[42]
LNCaP, DU145, and PC3 cells	Density ultracentrifugation	PKM2	protein	Therapeutic marker	Contributes to PCa metastasis to the bone	[98]
DU145 cells	Differential centrifugation	PCSEAT	lncRNA	Therapeutic marker	Plays an oncogenic role by contributing to growth and motility	[74]
Plasma	Total Exosome Isolation Reagent	SAP30L-AS1, SChLAP1	lncRNA	Diagnostic markers	Biochemical recurrence and disease progression	[76]
Urine	Ultracentrifugation	lincRNA-p21	lncRNA	Diagnostic marker	suppresses the progression of prostate cancer	[77]
Serum, PC3 cells	Differential centrifugation	P-glycoprotein	protein	Diagnostic marker	Contributes to docetaxel-resistance in PCa	[99]
Plasma	precipitation and ultracentrifugation	*KLK3*, *AR-V7*	mRNA	Diagnostic and prognostic markers	Contributes to disease progression	[100,101]
Serum, DU145 cells	ultracentrifugation	ACTN4	protein	Therapeutic marker	Contributes to disease progression and invasion	[45]

Abbreviations of mRNA and proteins are as follows—Transmembrane protein 256 (TM256), Late Endosomal/Lysosomal Adaptor, MAPK And MTOR Activator 1 (LAMTOR1),V-type proton ATPase 16 kDa proteolipid subunit (VATL), Adipogenesis regulatory factor (ADIRF), ETS Transcription Factor ERG (*ERG*), Prostate Cancer Antigen 3 (*PCA3*), Sterile alpha motif-pointed domain-containing Ets transcription factor (*SPDEF*), dimethylarginine dimethylaminohydrolase 1(DDAH1), Calsyntenin 1 (CLSTN1), Fatty acid synthase (FASN), Lactate Dehydrogenase A (LDHA), Annexin A2 (ANXA2), Clusterin (CLU), Enolase 1 (ENO1), Fibronectin 1 (FN1), Keratin 8 (KRT8), Laminin Subunit Alpha 5 (LAMA5), Nucleophosmin 1 (NPM1), Peroxiredoxin 1 (PRDX1), Transferrin Receptor (TFRC), Pyruvate kinase M2 (PKM2), PCa specific expression and *EZH2*-associated transcript (*PCSEAT),* SAP30L Antisense RNA 1 (Head To Head)(SAP30L-AS1), second chromosome locus associated with prostate-1 (SChLAP1), Kallikrein Related Peptidase 3 (*KLK3*), Androgen receptor splice variant 7 (*AR-V7*), Actinin Alpha 4 (ACTN4).

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
