# Peer review of "Role of Exosomes in Prostate Cancer Metastasis"

_ijms, 2021, doi:10.3390/ijms22073528_

Round 1

Reviewer 1 Report

The manuscript is well written and topical. Although the title of the manuscript is Role of exosomes in prostate cancer metastasis, the manuscript deals mostly with the structure and role of exosomes in general, while the data about the structure and role of exosomes in prostate cancer are in minority.  Also figure 1 represents exosome heterogeneity in general. Since the aim of the manuscript is to summarize and discuss the role of exosomes in prostate cancer I strongly suggest including a figure and/or table where all the information about the exosomes in prostate cancer are briefly and schematically shown and their potential positive or negative role or characteristics is summarized.

Also please reconsider including the data about exosome contained proteins in prostate cancer (section 2.1. – currently none specific is mentioned). Line 431 – question mark and a dot at the end of the sentence?.

Author Response

Comment 1: The manuscript is well written and topical. Although the title of the manuscript is Role of exosomes in prostate cancer metastasis, the manuscript deals mostly with the structure and role of exosomes in general, while the data about the structure and role of exosomes in prostate cancer are in minority.  Also figure 1 represents exosome heterogeneity in general.

Response: As suggested, we have now elaborated more on the data about the role of exosomes in prostate cancer. In addition, we have modified Section 2.1 to include studies on prostate cancer. While Fig. 1 represents exosome heterogeneity in general, we have included Table 1 that is focused on exosomal markers in prostate cancer.

Comment 2: Since the aim of the manuscript is to summarize and discuss the role of exosomes in prostate cancer. I strongly suggest including a figure and/or table where all the information about the exosomes in prostate cancer are briefly and schematically shown and their potential positive or negative role or characteristics is summarized.

Response: As suggested, we have now included a table summarizing the exosomal markers in prostate cancer. Please refer to Table 1 between lines 199 and 200 of the revised manuscript.

Comment 3: Also please reconsider including the data about exosome contained proteins in prostate cancer (section 2.1. – currently none specific is mentioned).

Response: As suggested, we have summarized data on exosome contained proteins in prostate cancer. Please refer to lines 91-112 of the revised manuscript.

Comment 4: Line 431 – question mark and a dot at the end of the sentence?.

Response: We have corrected this error in the revised manuscript.

Reviewer 2 Report

We know that prostate cancer is a type of cancer with poor blood supply, unlike kidney cancer. A considerable part of its metastasis route depends on the lymphatic system. This article discusses the role of exosomes in prostate cancer angiogenesis, but the role of exosomes in prostate cancer lymphatic metastasis and molecular mechanisms are not included. Perhaps adding this part of the content may make this article more systematic and comprehensive.

Author Response

Comment: We know that prostate cancer is a type of cancer with poor blood supply, unlike kidney cancer. A considerable part of its metastasis route depends on the lymphatic system. This article discusses the role of exosomes in prostate cancer angiogenesis, but the role of exosomes in prostate cancer lymphatic metastasis and molecular mechanisms are not included. Perhaps adding this part of the content may make this article more systematic and comprehensive.

Response: As suggested, we have included data on role of exosomes in prostate cancer lymphatic metastasis. Please refer to revised section 3.3 (lines 288-317) of the revised manuscript.
